# Tomographic Characterization of the European Shorthair Cat Orbital and Infraorbital Regions

**DOI:** 10.3390/ani16010147

**Published:** 2026-01-05

**Authors:** João Filipe Requicha, Ana Rita Sousa, Nuno Proença, Ana Válega, Sofia Alves-Pimenta

**Affiliations:** 1School of Agrarian and Veterinary Sciences, University of Trás-os-Montes and Alto Douro (UTAD), 5000-801 Vila Real, Portugal; 2Veterinary and Animal Science Research Centre (CECAV), Associate Laboratory for Animal and Veterinary Sciences (AL4AnimalS), University of Trás-os-Montes and Alto Douro (UTAD), 5000-801 Vila Real, Portugal; 3Hospital Veterinário do Porto, Onevet Group, Palmeiras 19, 4150-562 Porto, Portugal; 4Centre for the Research and Technology of Agro-Environmental and Biological Sciences (CITAB) Inov4Agro, University of Trás-os-Montes and Alto Douro (UTAD), 5000-801 Vila Real, Portugal

**Keywords:** computed tomography, feline anatomy, dentistry, morphometry, trigeminal nerve, infraorbital foramen, locoregional analgesia

## Abstract

This study describes the European Shorthair cat’s orbital and infraorbital anatomy to support safer locoregional anesthesia and surgical planning. The objective was to provide reliable anatomical landmarks and reference values, using an approach based on linear measurements taken directly from sagittal, transverse and dorsal CT planes, instead of three-dimensional reconstructions. By analyzing scans from 24 neutered cats, we confirmed that those measurements can be consistently repeated. Mean infraorbital canal length was 5.23 ± 0.49 mm. Significant correlations were observed between infraorbital and skull parameters, which may offer approximate estimations to assist clinical decision-making. After adjustment for skull size, some significant differences between sexes remained.

## 1. Introduction

The orbital and infraorbital regions and their relation to the pterygopalatine fossa in the domestic cat have significant clinical importance in veterinary dentistry and maxillofacial surgery, and ophthalmology [1,2]. The maxillary nerve (branch of trigeminal nerve) exits the skull through the foramen rotundum and exteriorizes in the rostral alar foramen of the sphenoid. Courses ventrally to eyeball in the pterygopalatine fossa, to finally enter the maxillary foramen. Along with its branch, the infraorbital nerve, which traverses the infraorbital canal, emerging through the infraorbital foramen, supplies sensory innervation to the maxillary teeth, upper lip, and surrounding structures. In cats, the infraorbital canal is much shorter than the bony orbit floor, and the risk of globe penetration is higher [3]. Accurate localization of these landmarks is essential for safe infraorbital nerve blocks and for planning surgical approaches [4]. However, detailed morphometric descriptions of this region remain scarce, limiting the availability of reference values for clinical application.

In humans, several cadaveric and imaging studies have described in detail the infraorbital region, foramen, canal and nerve, as well as their relationships to adjacent structures as the maxillary teeth, and palpable soft tissue landmarks to aid clinical localization [5,6,7,8,9,10]. Agthong et al. (2005) measured distances from the infraorbital foramen to midline, showing sex-related differences [11]. Accessory foramina were described [5,7,8]. Infraorbital nerve was reported as a relevant landmark for the pterygopalatine fossa, cavernous sinus, and anterolateral skull base in neurosurgery [12]. Collectively, these studies provide a robust anatomical framework but are limited to humans.

In veterinary medicine, work is scarcer. In dogs, anesthesia techniques have been studied, with an ultrasound-guided peribulbar block validated in cadavers by Foster and colleagues (2021) and retrobulbar versus peribulbar techniques compared clinically [1,13]. Winer et al. (2018) described retrobulbar disease using computed tomography (CT), illustrating the importance of imaging in orbital diagnostics [14]. These works emphasize that infraorbital and orbital morphometry directly influence anesthesia safety and effectiveness across species.

In cats, Davis et al. (2021) performed a CT-based analysis of the infraorbital foramen and canal, relying on multiplanar and 3D reconstructions to describe morphometry and skull-type differences [15]. Ramos et al. (2021) characterized a different cohort of European Shorthair skulls morphometrically using CT, though without focusing on the infraorbital region [16]. Cadaveric research has explored peribulbar and retrobulbar nerve blocks in cats [17], and more recently, a percutaneous [18], a transpalpebral approach [19] and an ultrasound-guided trigeminal approach at the pterygopalatine fossa were described [20], highlighting the need for accurate morphometric guidance.

The aims of this study were to analyze retrospectively clinical scans from European Shorthair Cats, using a simplified approach based on linear measurements taken directly from sagittal, transverse and dorsal CT planes, instead of 3D reconstructions previously described [15], to assess repeatability, report infraorbital and skull ratios, and develop regression models to predict infraorbital dimensions from basic skull measures. By proposing this simplified and clinic-ready workflow, our study contributes practical reference data and provides a foundation for standardization of feline infraorbital morphometry.

## 2. Materials and Methods

This retrospective study analyzed head CT examinations from domestic cats observed at the Onevet Hospital Veterinário Porto (Portugal). The selected CT images were retrospectively evaluated, and no animal was used or handled for the purpose of this study. European Shorthair cats older than 24 months were included to ensure skeletal maturity [21,22]. Exclusion criteria were cranial abnormalities or poor image quality. Twenty-four CT scans of neutered feline heads, of both sexes (12 females and 12 males), aged between 4 and 17 years (mean age ± standard deviation (SD) of 10.14 ± 3.73 years) were evaluated. Their mean weight was 4.67 ± 1.31 kg, ranging from 3.25 kg to 7.65 kg.

Regarding the imaging protocol, CT scans were performed with a 16-slice helical scanner (Aquilion Prime SP, Canon, Otawara, Japan). Cats were positioned in ventral recumbency with their head aligned and immobilized. Transverse slices of the head were acquired at ~120 kVp, 120 mAs, with a slice thickness of 1 mm, and a slice interval of 0.8 mm. Data was stored in the Digital Imaging and Communications in Medicine (DICOM) format.

In order to perform the linear measurements described in Table 1, firstly, rotational misalignment of the head was corrected by aligning the images in the three planes, as demonstrated in Figure 1. The sagittal plane (Figure 1A) was aligned at the level of the nasal septum and the interincisive midline; the transverse plane (Figure 1B) was oriented perpendicular to the hard palate; and the dorsal plane (Figure 1C) was parallel to the hard palate. Since the computer program used allowed for the simultaneous visualization of the three anatomical planes (sagittal, transverse, and dorsal), the identification of the measurement planes and the delimitation of the anatomical structures to be measured was performed with greater precision.

Linear morphometric parameters (Table 1) were obtained directly from CT images using Horos^TM^ v3.3.6.dmg DICOM software with bone filters. Prior to measurement, all images were calibrated from pixels to millimeters. A preliminary study and prior training were conducted, to establish anatomical landmarks that would allow for the repeatability of the measurement methodology by J.F.R., A.R.S., S.A.-P. Recorded variables included infraorbital foramen major axis (Figure 2), minor axis (Figure 3) and length (Figure 4), distance between infraorbital foramina (DIF) (Figure 5), orbital height and width (Figure 6 and Figure 7), zygomatic arch width (Figure 8), skull width and length (Figure 9). Ratios were calculated to normalize for skull size. As preliminary results revealed adequate repeatability of measurements, and in order to reduce the analysis margin of error, two measurements of each studied parameter were performed. The measurements were performed by the same operator (to reduce interpersonal errors), and each measurement of each parameter was performed at different times, in order to reduce intrapersonal errors. Then, the arithmetic mean of the measurements was calculated.

Data were summarized using descriptive statistics analysis and analyzed using a computer software system (SPSS Statistics for Windows version 23.0; IBM Corp., Armonk, NY, USA). Significance was set at *p* < 0.05. The normality of data was assessed using the Shapiro–Wilk test.

Intra-observer repeatability was assessed using Bland–Altman analysis and intraclass correlation coefficients (ICC) [23,24]. In the Bland–Altman analysis, the 95% limits of agreement (LA) were calculated as the mean difference ± 1.96 SD, where SD is the standard deviation of all individual differences. When the 95% confidence interval (CI) of the mean differences includes the 0, measurements are considered to be in agreement; and when the 95% lower and upper limits of agreement are small, measurements are considered to be equivalent, with adequate intra-observer agreement [23]. An ICC of 1.0 indicates perfect agreement, and an ICC of 0.0 indicates no agreement [24,25]. Pearson correlations explored associations between variables, and linear regression analysis were applied when adequate. The sex and side differences were evaluated using the paired sample *t*-test. Effect sizes (Cohen’s *d*) were calculated to quantify the magnitude of sex differences.

## 3. Results

The repeatability analysis showed that all mean differences (*d* values) approached zero. The highest *d* value found by the Bland–Altman method was obtained in the comparison between session 1 and session 2 of measurements of the skull width parameter (*d* = 0.13) (Table 2). All the 95% CI of the mean differences included the 0, meaning that measurements are considered to be in agreement; and the intervals between 95% lower and upper limits of agreement are small, so, measurements are considered to be equivalent, with adequate intra-observer agreement [23]. Regarding the ICC, lower limits of the 95% CI ≥ 0.75 indicate a strong association between the two measurements [24,25]. The ICC for intraobserver repeatability were all ≥0.75, with the lowest value (0.76) observed for orbital width (OW), indicating statistically adequate repeatability.

Descriptive statistical analysis yielded the values presented in Table 3 for the evaluated morphometric parameters and the values presented in Table 4 for the calculated indices. Mean values and standard deviations were obtained for all morphometric parameters and calculated ratios. No differences were found between right and left sides.

Pearson correlation measures the strength and direction of linear relationships between variables, ranging from −1.0 to +1.0, where weak (0.0 to 0.3), moderate (0.3 to 0.7), and strong (0.7 to 1.0) correlations can be found. A moderate positive correlation was found between skull width (SW) and infraorbital canal length (ICL) (*r* = 0.476, *p* = 0.001). From the linear regression analysis between the ICL and the SW we obtained the regression equation: *y* = 0.0739*x* + 0.7427, where *y* = ICL and *x* = SW (*r*^2^ = 0.227).

Also, a moderate positive correlation was observed between skull length (SL) and infraorbital foramen minor axis (IFmA) (*r* = 0.531, *p* < 0.001). From the linear regression analysis between the SL and the IFmA we obtained the regression equation: *y* = 0.0499*x* − 1.772, where *y* = IFmA and *x* = SL (*r*^2^ = 0.282).

Significant differences were observed between sexes (Table 5 and Table 6). Males exhibited significantly larger dimensions than females in multiple parameters, including infraorbital foramen major (IFMA) and minor axis (IFmA), distance between infraorbital foramina (DIF), orbital height (OH), infraorbital canal length (ICL), skull length (SL), skull width (SW), and several ratios. When corrected for skull size using ratios, some of these sex differences were no longer significant. However, some ratios kept differences between sex, as Infraorbital minor axis/orbital width (IFmA/OW), Infraorbital minor axis/zygomatic arch width (IFmA/ZAW), distance between infraorbital foramina (IFD)/skull width (IFD/SW), and inclusively skull width/skull length (SW/SL).

## 4. Discussion

This study provides reference morphometric data for the infraorbital region of European Shorthair cats while critically evaluating the use of simplified CT-based measurements in feline craniofacial anatomy. Unlike previous studies that predominantly relied on multiplanar reformatting or three-dimensional reconstructions, the present work demonstrates that linear measurements obtained directly from standard sagittal, transverse, and dorsal CT planes are sufficient to characterize clinically relevant infraorbital and orbital landmarks. The morphometric values reported here are largely consistent with those described in earlier CT-based investigations, although minor differences were observed, likely reflecting methodological variation, skull conformation, and sample characteristics [15]. These findings suggest that increased methodological complexity does not necessarily confer greater anatomical precision and underscore the importance of balancing accuracy with clinical feasibility when translating imaging-based morphometry into routine veterinary practice.

In our retrospective analysis of clinical scans, intraobserver repeatability was statistically adequate in the two measurement sessions for each morphometric parameter studied, so we consider that, in the future, for a trained observer, only one measurement session should be considered, rather than the two sessions performed in this study [23,24,25]. To minimize the operator-dependent measurement variability, a preliminary study was conducted, to establish anatomical landmarks that would al-low for the repeatability of the measurement methodology by J.F.R., A.R.S., S.A.-P. In the measurements performed, the largest standard deviation of the differences was related to the skull length (SL) measurement, and the smallest corresponded to the infraorbital foramen minor axis (IFmA) measurement (Table 2). These standard deviation values are proportional to the magnitude of the dimensions under consideration.

We found the major axis of the infraorbital foramen to have an oblique orientation relative to the mid-sagittal plane (Figure 2), unlike what has been described in dogs, and, for that reason, naming it total vertical height of the infraorbital foramen seemed inaccurate to us, so we rather named it the infraorbital foramen major axis (IFMA) and the infraorbital foramen minor axis (IFmA) [2].

There is an increasing demand for reducing iatrogenic injuries during local anesthetic procedures. Therefore, accurate measurements should be considered when creating anesthetic protocols for maxillary and infraorbital nerve blocks. The results obtained are in line with the values described by [15], who conducted a study of the infraorbital foramen in cat skulls using three-dimensional CT reconstructions in mesocephalic cats (such as the Common European breed), although this approximation being always by default. This can be explained by the technique used, since three-dimensional reconstructions eliminate the penumbra that exists in two-dimensional CT images and which we considered in the measurements performed on multiplanar slices. Also, in the study from their team, the mean age was superior to ours (15.5 years). The authors also evaluated brachycephalic cats, which had greater infraorbital foramen heights but considerably shorter infraorbital foramen lengths than the mesocephalic cats evaluated in our study. Their findings demonstrate that there are important variations in the individual anatomic size, shape, and positioning of the infraorbital foramen and infraorbital canal in cats [15]. In that regard, our results contribute to establish normal values in neutered European Shorthair cat.

Ratios between measurements were established, as described in Table 4. Using ratios of the measurements acquired allows comparisons between values from animals of different sizes, eliminating the effect of skull size, and potential clinical algorithms, which may be useful in clinical procedures and anesthetic nerve blocks. In a preliminary analysis, we observed that skull length (SL) values (91.55 ± 5.68 mm) were ~20 superior to infraorbital foramen major axis (IFMA) values (3.82 ± 0.68 mm), so we decided to multiply by 20 the IFMA to simplify the interpretation. Regarding the ratio (infraorbital foramen minor axis × 2)/skull width [(IFmA × 2)/SW], we multiplied by two the IFmA, as SW includes the left and right IFmA. Anesthetic nerve blocks represent an effective complement to general anesthesia in current multimodal analgesic regimes, as they reduce the required drug dose and their systemic effects, reduce the need for inhaled and intravenous anesthetics, and provide effective postoperative analgesia [26]. Because the infraorbital nerve supplies sensory fibers to the maxillary premolar, canine, and incisor teeth, it is extremely useful to block it in stomatology surgery procedures. Also, the maxillary nerve supplies maxillary molar area and dents [19]. Routinely, the infraorbital foramen is located by palpation between the dorsal edge of the zygomatic process and the gingiva of the canine tooth, and to effectively block the infraorbital nerve, some authors defend the anesthetic must be injected into the most caudal portion of the infraorbital canal [19]. Prior cadaveric experimental works shown that dye deposition at the foramen is effective while also avoiding complications such as ocular trauma associated with catheter placement [18] and dye migration intracranially via the alar foramina [13]. In fact, caution is warranted when performing peribulbar or infraorbital injections, as canine cadaver studies have demonstrated the potential for injectate to migrate intracranially via the alar foramina [13].

Regression models to predict infraorbital dimensions from basic skull measures were developed. To be able to accurately infer the dimensions of the infraorbital orifice and the infraorbital canal length without resorting to advanced imaging techniques, we acquired the length and width of the animal’s skull, which are easily obtained. To this end, we calculated the correlation coefficients and found a significant correlation between the infraorbital canal length (ICL) and skull width (SW) of 0.476, and between the infraorbital foramen minor axis (IFmA) and skull length (SL) of 0.531. From the regression analysis between the ICL and the SW, we obtained the regression line: *y* = 0.0739*x* + 0.7427; and from the regression analysis between the infraorbital foramen minor axis (IFmA) and skull length (SL), the regression equation: *y* = 0.0499*x* − 1.772. Although the correlations between infraorbital canal length (ICL) and skull width (SW), and between infraorbital foramen minor axis (IFmA) and skull length (SL), were statistically significant, their magnitude was moderate (*r* ≈ 0.47–0.53). This indicates that these regression models explain only a limited proportion of the observed anatomical variability and should therefore be interpreted with caution. Consequently, the proposed equations are not intended to provide precise predictions at an individual level, but rather to offer approximate estimations that may assist clinical decision-making when advanced imaging is unavailable.

Our measurements underscore anatomical variability between sexes which may influence clinical outcomes. Significant differences were found between the sexes, particularly evident in the infraorbital foramen major axis (IFMA), infraorbital foramen minor axis (IFmA), skull length (SL) and skull width (SW). After obtaining the ratios, some parameters no longer shown a significant difference between males and females, reflecting that the differences are proportional to the differences in the dimensions of the skull, fact that is consistent with the sexual dimorphism described in feline species [27,28], including domestic cats [16,29]. However, some ratios kept differences between sex, as infraorbital foramen minor axis/orbital width (IFmA/OW), infraorbital foramen minor axis/zygomatic arch width (IFmA/ZAW), distance between infraorbital foramina (IFD)/skull width (IFD/SW), and inclusively skull width/skull length (SW/SL), being the reflect of a skull proportionally longer than wide in males, as previously described in the literature [16].

Globe penetration following maxillary nerve block for dental surgery in cats is one of the major risks of the procedure [4]. Previous authors related that the amount of fat present in the ventral floor of the orbit may influence clinical outcomes [18]. The transversal images defined as described in the methodology of this study may be useful for measuring the amount of orbital fat with specific anatomical landmarks. Nevertheless, the percutaneous approach (also known as the subzygomatic) described in cat cadavers, by inserting a needle percutaneously between the rostroventral zygomatic arch and rostral border of the coronoid process of the mandible orienting medio-rostrally toward the estimated location of the maxillary foramen, may be associated with higher risks of globe penetration, attending to the anatomical structures involved, so we do not recommend as routine access to be performed in living cats [18]. Furthermore, awareness of the approximate ICL can guide clinicians during nerve blocks: advancing the needle only a few millimeters into the canal (often no more than the length of the canal itself) reduces the likelihood of entering the orbital space. Combined with gentle technique, palpation of anatomical landmarks, and, when available, image guidance, this knowledge may help reduce the risk of globe penetration and other iatrogenic complications.

The available literature addressing the orbital and infraorbital regions in the cat remains limited, particularly with respect to detailed morphometric characterization based on imaging studies. As a result, reference data for this anatomical area are still scarce, which may constrain direct comparisons across studies. Within this context, despite the relatively small sample size (*n* = 24) and the non-random clinical scans study, the present work contributes additional imaging-based morphometric information for mesocephalic cats and complements previous descriptions of feline cranial anatomy, providing a basis for future comparative and clinically oriented investigations. Standardizing the technique implemented in this study would be beneficial so that it can be performed with greater accuracy and ease. Other applications, such as those described in human neurosurgery, may be explored in the future [12].

Overall, this study contributes imaging-based anatomical reference data for the feline infraorbital region and demonstrates that simplified CT-derived measurements can provide clinically relevant information, supporting safer and more informed application of locoregional anesthetic and surgical techniques in cats.

## 5. Conclusions

This study provides CT-based reference values for the infraorbital region of neutered European Shorthair cats using a simplified, repeatable measurement approach. The findings confirm that the infraorbital canal is short and lies in close proximity to the orbit, underscoring the anatomical relevance of this region for infraorbital and maxillary nerve block techniques.

From an anatomical perspective, these results suggest that deep needle or catheter advancement into the infraorbital canal may not be necessary to achieve effective regional anesthesia and may increase the risk of iatrogenic injury. However, this interpretation is based on morphometric data rather than prospective clinical outcome studies and should therefore be applied with appropriate caution.

The simplified methodology proposed may facilitate broader clinical use of CT-derived morphometry without the need for advanced image reconstruction techniques. Nevertheless, the results are limited to mesocephalic European Shorthair cats, and extrapolation to other breeds or skull conformations should be undertaken carefully. Future studies incorporating larger and more diverse feline populations, as well as prospective clinical outcome data, are warranted to further refine anatomical guidelines for safer locoregional anesthetic techniques.

## Figures and Tables

**Figure 1 animals-16-00147-f001:**
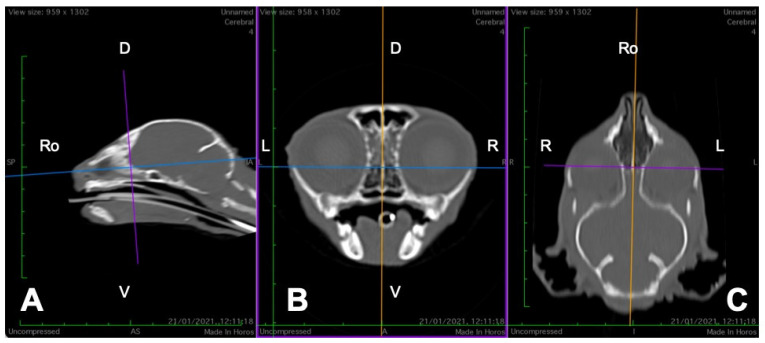
Representative multiplanar computed tomography reconstruction showing the sagittal (**A**), transverse (**B**), and dorsal (**C**) planes. The sagittal plane was aligned at the level of the nasal septum and the interincisive midline; the transverse plane was oriented perpendicular to the hard palate; and the dorsal plane was parallel to the hard palate. D, dorsal, L, left, R, right, Ro, rostral, V, ventral.

**Figure 2 animals-16-00147-f002:**
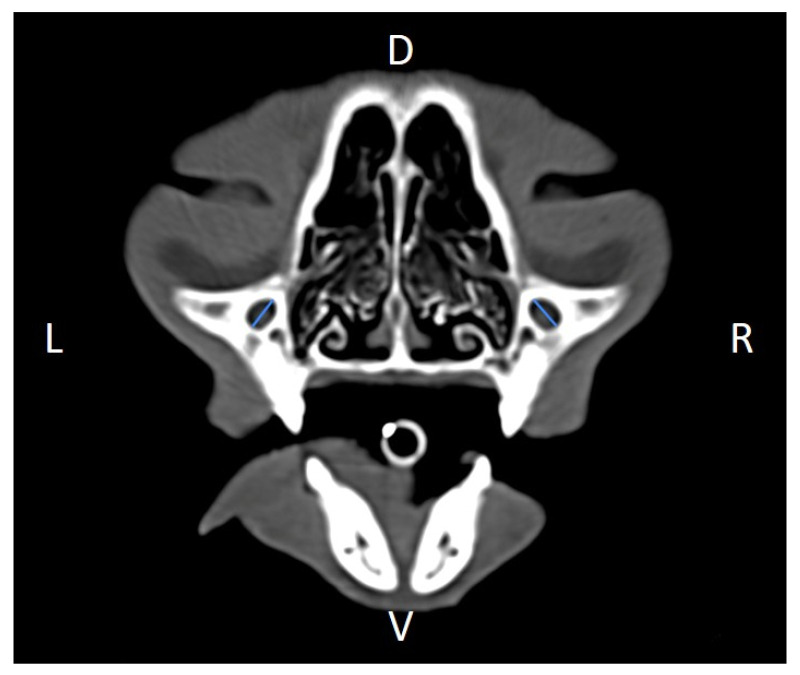
Infraorbital major axis (IFMA) measured at the left and right sides, on a transverse CT image (blue lines). D, dorsal, L, left, R, right, V, ventral.

**Figure 3 animals-16-00147-f003:**
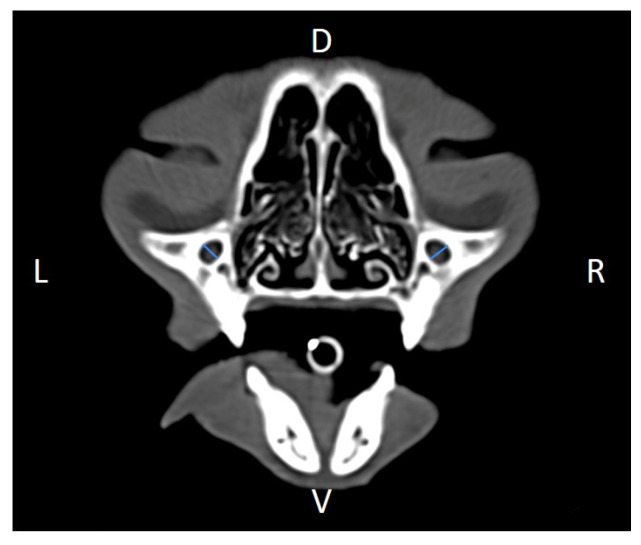
Infraorbital minor axis (IFmA) measured at the left and right sides, on a transverse CT image (blue lines). D, dorsal, L, left, R, right, V, ventral.

**Figure 4 animals-16-00147-f004:**
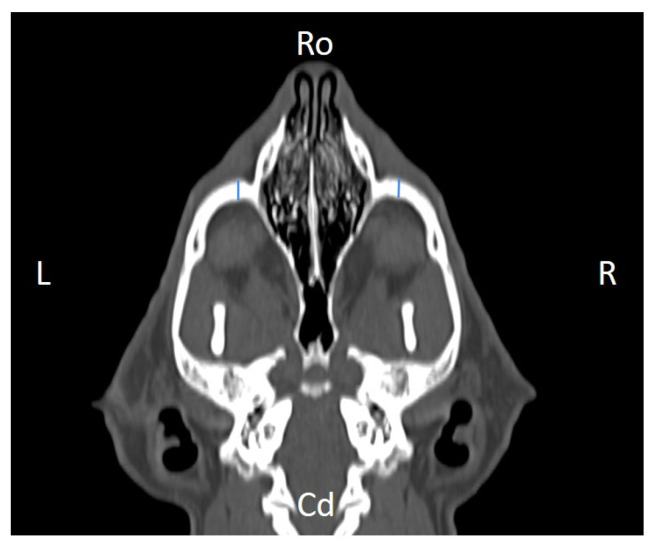
Infraorbital canal length (ICL) measured at the left and right sides, on a dorsal CT image (blue lines). Cd, caudal, L, left, R, right, Ro, rostral.

**Figure 5 animals-16-00147-f005:**
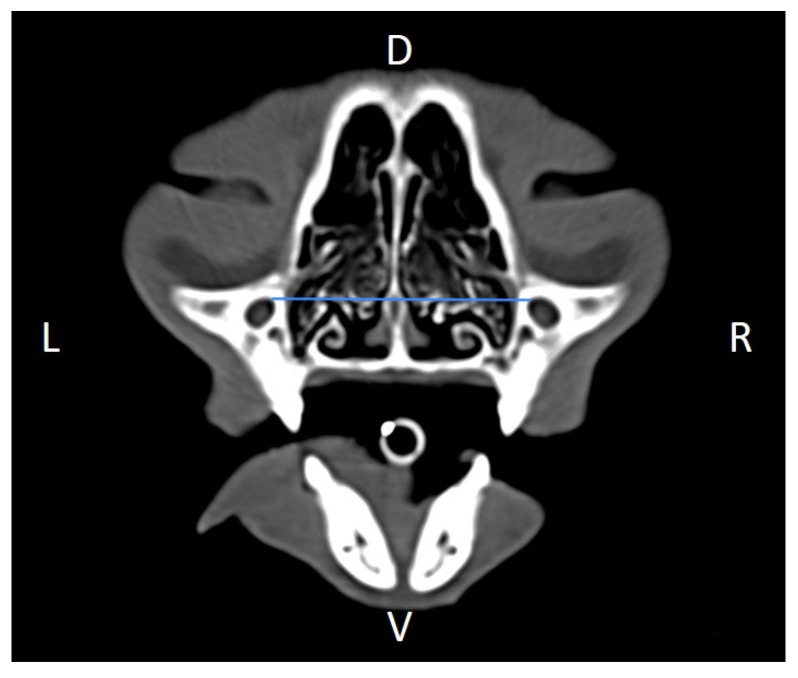
Distance between infraorbital foramina (DIF) measured on a transverse CT image (blue line). D, dorsal, L, left, R, right, V, ventral.

**Figure 6 animals-16-00147-f006:**
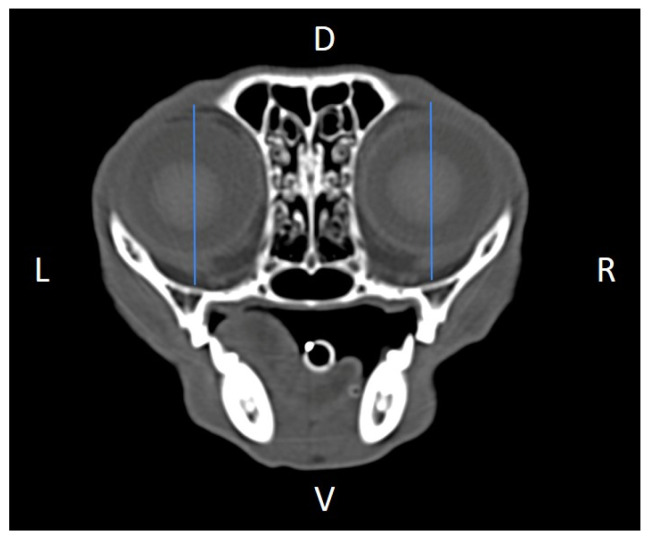
Orbital height (OH) measured at the left and right sides, on a transverse CT image (blue lines). D, dorsal, L, left, R, right, V, ventral.

**Figure 7 animals-16-00147-f007:**
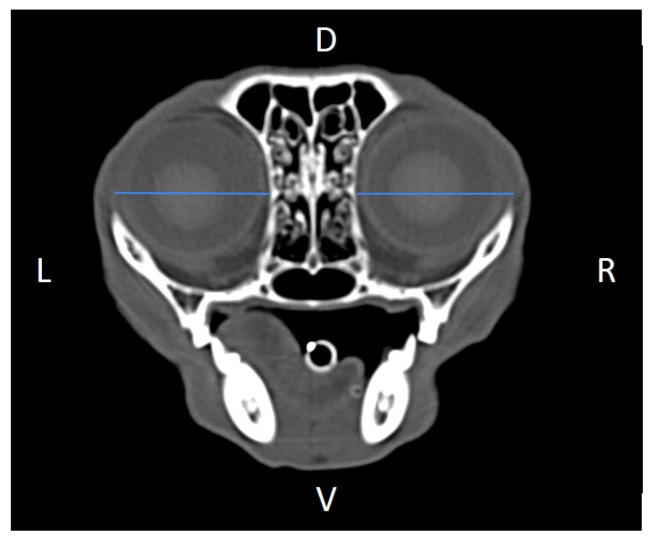
Orbital width (OW) measured at the left and right sides, on a transverse CT image (blue lines). D, dorsal, L, left, R, right, V, ventral.

**Figure 8 animals-16-00147-f008:**
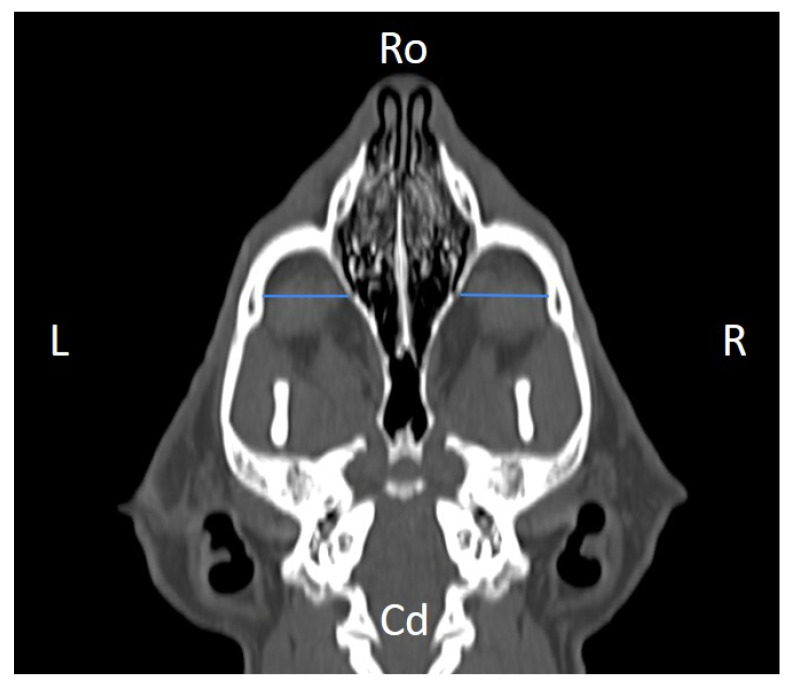
Zygomatic arch width (ZAW) measured at the left and right sides, on a dorsal CT image (blue lines). Cd, caudal, L, left, R, right, Ro, rostral.

**Figure 9 animals-16-00147-f009:**
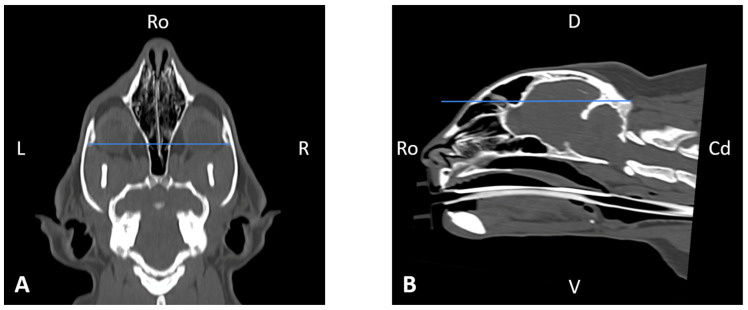
Skull width (SW) measured on a dorsal CT image (**A**) (blue line), and skull length (SL) measured on a sagittal CT image (**B**) (blue line). Cd, caudal, D, dorsal, L, left, R, right, Ro, rostral, V, ventral.

**Table 1 animals-16-00147-t001:** Description of the morphometric parameters assessed and the anatomical landmarks used in the study.

Anatomical Region	Morphometric Parameter (Definition)
**Infraorbital foramen and canal**	**Infraorbital major axis (IFMA):** largest diameter of the infraorbital foramen, the longest straight line connecting opposite points of the infraorbital foramen through its center.
	**Infraorbital minor axis (IFmA):** smallest diameter of the infraorbital foramen, the shortest straight line connecting opposite points of the infraorbital foramen through its center, and perpendicular to the major axis.
	**Infraorbital canal length (ICL):** distance between the maxillary foramen and the infraorbital foramen.
	**Distance between infraorbital foramina (DIF):** distance between the two most medial points of the left and right infraorbital foramina.
**Orbit**	**Orbital height (OH):** maximum distance from the orbital floor to the orbital roof.
	**Orbital width (OW):** maximum distance between the most medial and most lateral points of the orbit.
**Zygomatic arch**	**Zygomatic arch width (ZAW):** maximum internal width of the zygomatic arch, measured on the slice at the level of the infraorbital canal, rostral to the frontal process of the zygomatic bone.
**Skull**	**Skull width (SW):** maximum width between the zygomatic arches, measured at the base of the frontal process of the zygomatic bone.
	**Skull length (SL):** distance from the external occipital protuberance to a perpendicular dropped to the rostral limit of the nasal bone.

**Table 2 animals-16-00147-t002:** Intraclass correlation coefficients (ICC), mean differences, and limits of agreement (LA) obtained on the parameters evaluated.

Parameter	ICC (95% CI)	*d* ± SEM (mm)	*d* 95% CI (mm)	*d* ± SD (mm)	95% LA (mm)
Infraorbital foramen major axis (IFMA)	0.91 (0.85–0.95)	−0.051 ± 0.041	−0.133 to 0.032	−0.051 ± 0.284	−0.607 to 0.506
Infraorbital foramen minor axis (IFmA)	0.89 (0.81–0.94)	−0.022 ± 0.032	−0.087 to 0.043	−0.022 ± 0.223	−0.460 to 0.416
Distance between infraorbital foramina (DIF)	0.98 (0.96–0.99)	0.000 ± 0.070	−0.145 to 0.145	0.000 ± 0.344	−0.674 to 0.674
Orbital height (OH)	0.89 (0.81–0.94)	−0.007 ± 0.094	−0.197 to 0.183	−0.007 ± 0.647	−1.276 to 1.262
Orbital width (OW)	0.86 (0.76–092)	0.031 ± 0.081	−0.132 to 0.195	0.031 ± 0.557	−1.061 to 1.123
Infraorbital canal length (ICL)	0.89 (0.81–0.94)	0.044 ± 0.034	−0.025 to 0.113	0.044 ± 0.238	−0.422 to 0.510
Zygomatic arch width (ZAW)	0.93 (0.87–0.96)	−0.069 ± 0.132	−0.335 to 0.197	−0.069 ± 0.915	−1.863 to 1.725
Skull length (SL)	0.97 (0.92–0.99)	−0.081 ± 0.275	−0.651 to 0.489	−0.081 ± 1.318	−2.665 to 2.503
Skull width (SW)	0.99 (0.98–1.00)	0.132 ± 0.096	−0.067 to 0.331	0.132 ± 0.471	−0.792 to 1.055

CI, confidence interval; *d*, mean difference; SEM, standard error of the mean; SD, standard deviation.

**Table 3 animals-16-00147-t003:** Results of the descriptive statistics of the morphometric parameters assessed in the 24 adult European Shorthair cats.

Parameter	Mean ± SD (range) (mm)
Infraorbital foramen major axis (IFMA)	3.82 ± 0.68 (2.56–5.58)
Infraorbital foramen minor axis (IFmA)	2.78 ± 0.47 (1.97–3.88)
Distance between infraorbital foramina (DIF)	27.66 ± 1.88 (24.00–30.60)
Orbital height (OH)	25.52 ± 1.54 (22.20–28.80)
Orbital width (OW)	23.46 ± 1.11 (20.07–25.40)
Infraorbital canal length (ICL)	5.23 ± 0.49 (4.42–6.33)
Zygomatic arch width (ZAW)	14.08 ± 2.67 (10.10–19.70)
Skull length (SL)	91.55 ± 5.68 (81.30–110.00)
Skull width (SW)	60.80 ± 3.15 (53.39–66.16)

SD, standard deviation.

**Table 4 animals-16-00147-t004:** Results of the calculated ratios for the 24 adult European Shorthair cats.

Ratio	Mean ± SD (range)
Infraorbital foramen major axis/Orbital height (IFMA/OH)	0.15 ± 0.03 (0.1–0.23)
(Infraorbital foramen major axis × 20)/Skull length [(IFMA × 20)/SL]	0.84 ± 0.15 (0.56–1.23)
(Infraorbital foramen minor axis × 2)/Skull width [(IFmA × 2)/SW]	0.18 ± 0.07 (0.1–0.33)
Infraorbital foramen minor axis/Orbital width (IFmA/OW)	0.12 ± 0.02 (0.09–0.17)
Infraorbital foramen minor axis/Zygomatic arch width (IFmA/ZAW)	3.91 ± 0.67 (2.85–6.27)
Distance between infraorbital foramina/Skull width (DIF/SW)	0.46 ± 0.03 (0.41–0.50)
Orbital height/Skull length (OH/SL)	0.28 ± 0.02 (0.26–0.33)
Orbital width/Skull length (OW/SL)	0.39 ± 0.03 (0.34–0.47)
Orbital width/Zygomatic arch width (OW/ZAW)	1.69 ± 0.27 (1.17–2.22)
(Infraorbital canal length × 20)/Skull length [(ICL × 20)/SL]	1.15 ± 0.10 (0.94–1.35)
Zygomatic arch width/Skull width (ZAW/SW)	0.24 ± 0.04 (0.17–0.32)
Skull width/Skull length (SW/SL)	0.67 ± 0.04 (0.60–0.73)

SD, standard deviation.

**Table 5 animals-16-00147-t005:** Values and mean differences observed between males and females in the assessed morphometric parameters in mm.

Parameter	Sex	Mean ± SD (range) (mm)	Mean Differences (SEM)	*p* Value	Cohen’s *d*
Infraorbital foramen major axis (IFMA)	F	3.61 ± 0.66 (2.55–5.20)	−0.418 (0.190)	0.038	−0.576
	M	4.03 ± 0.65 (2.72–5.58)			
Infraorbital foramen minor axis (IFmA)	F	2.60 ± 0.34 (1.97–3.20)	−0.358 (0.142)	0.019	−0.515
	M	2.96 ± 0.52 (2.04–3.88)			
Distance between infraorbital foramina (DIF)	F	26.44 ± 1.54 (24.00–28.60)	−2.508 (0.668)	0.003	−1.084
	M	28.95 ± 1.24 (26.40–30.60)			
Orbital height (OH)	F	24.98 ± 1.68 (22.30–28.70)	−1.078 (0.339)	0.004	−0.664
	M	26.06 ± 1.23 (24.40–28.80)			
Infraorbital canal length (ICL)	F	5.02 ± 0.42 (4.42–5.74)	−0.445 (0.121)	0.001	−0.753
	M	5.46 ± 0.46 (4.50–6.33)			
Skull length (SL)	F	88.06 ± 3.16 (81.30–91.80)	−6.481 (1.388)	<0.001	−0.995
	M	94.54 ± 4.52 (87.90–100.01)			

F, female; M, male; SD, standard deviation; SEM, standard error of the mean.

**Table 6 animals-16-00147-t006:** Values and mean differences observed between males and females in the calculated ratios.

Parameter	Sex	Mean ± SD (range)	Mean Differences (SEM)	*p* Value	Cohen’s *d*
Infraorbital foramen minor axis/Orbital width (IFmA/OW)	F	0.11 ± 0.01 (0.09–0.14)	−0.011 (0.005)	0.039	−0.459
	M	0.12 ± 0.02 (0.09–0.17)			
Infraorbital foramen minor axis/Zygomatic arch width (IFmA/ZAW)	F	3.68 ± 0.62 (2.85–5.37)	−0.475 (0.199)	0.025	−0.489
	M	4.15 ± 0.64 (3.13–6.27)			
Distance between infraorbital foramina/Skull width (DIF/SW)	F	0.44 ± 0.02 (0.41–0.50)	−0.029 (0.006)	0.001	−1.309
	M	0.471 ± 0.02 (0.44–0.50)			
Skull width/Skull length (SW/SL)	F	0.69 ± 0.02 (0.66–0.71)	0.036 (0.010)	0.001	0.788
	M	0.65 ± 0.04 (0.60–0.73)			

F, female; M, male; SD, standard deviation; SEM, standard error of the mean.

## Data Availability

The raw data supporting the conclusions of this article will be made available by the authors on request.

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
