# Peer review of "Tomographic Characterization of the European Shorthair Cat Orbital and Infraorbital Regions"

_animals, 2026, doi:10.3390/ani16010147_

Round 1

Reviewer 1 Report

Comments and Suggestions for Authors

This manuscript provides clinically relevant morphometric data for the orbital and infraorbital regions of European Shorthair cats using linear CT measurements. The topic is important for veterinary dentistry, ophthalmology, and anesthesiology. The methodological approach is sound, and the data are useful; however, several sections require clarification, improved structure, additional explanation, and refinement of language. Numerous sentences can be simplified, and some results and interpretations should be expanded or clarified.

I have included page- and line-specific comments below, and additional detailed PDF annotations have also been inserted to assist with revision.

 SECTION-WISE COMMENTS

  1. Simple Summary (Page 1)

P1 – Lines 3–7 (“This study describes the European Shorthair Cats orbital…”)

Please simplify long sentences for readability. The current text is dense.

(See PDF annotation.)

P1 – Line 10 (“using simplified computed tomographic measurements”)

Clarify what is meant by “simplified”—simplified compared to which previous CT protocols? (PDF annotation included.)

  1. Abstract (Page 1)

P1 – Lines 17–19 The sentence beginning “Despite its clinical importance…” repeats content from the Simple Summary. Consider condensing.

P1 – Line 26 When reporting “Mean infraorbital canal length was 5.23 mm…”, please briefly mention its clinical significance (risk of globe penetration, block depth, etc.).(See PDF note.)

P1 – Lines 29–34

The sentence on sex differences is lengthy; consider restructuring for clarity.

  1. Introduction (Pages 2–3)

P2 – Lines 6–20 The anatomy description is lengthy. Consider condensing and focusing more on the gap in feline morphometry.(PDF annotations highlight suggested trimming.)

P2 – Lines 21–42 – Human morphometry section

This section is useful but too long; condense to the essential points. Emphasize relevance to feline anatomy.

P3 – Line 9 (“The aims of this study were…”) Please explicitly state the novelty:

— Linear-plane measurements directly from CT planes vs.

— 3D reconstructions in previous studies (e.g., Davis et al.).(PDF comment added.)

  1. Materials and Methods (Pages 4–5)

P4 – Line 4 (“older than 9 months”)

Provide justification or reference for skeletal maturity at 9 months.

P4 – Lines 12–15 (“Images were acquired at ~120 kVp…”)

Specify slice interval (spacing), not just thickness.

P4 – Lines 20–30 (“The sagittal plane… dorsal plane…”)

Clarify how rotational misalignment of the head was corrected before measurements. This is important for reproducibility.

P4 – Line 33 (“two measurements of each parameter were performed”)

Justify why two measurements were chosen instead of three (common in morphometric studies).(PDF note included.)

P5 – Figures 2–5

Add orientation labels (L/R, dorsal/ventral) to improve interpretation.

(Annotated on PDF.)

  1. Results (Pages 7–11)

P7 – Line 3 (“Twenty-four European Shorthair cats…”)

Include neuter status if available. Sex hormones can influence cranial morphology.

P7 – Lines 12–15 (“repeatability analysis…”)

Explain what ICC ≥ 0.76 indicates (good to excellent repeatability).

(PDF annotation added.)

P7 – Lines 18–30 (Correlation and regression)

Please report R² values in addition to r values to quantify predictive strength.

Tables 2–4 (Pages 8–9)

Some ratio formulas (e.g., IFMA × 20) are unclear. Please explain the rationale for numerical multipliers.

  1. 6. Discussion (Pages 12–14)

P12 – Lines 1–8 (Opening paragraph)

This paragraph restates methodological details; consider focusing more on interpretation and comparison.

P12 – Lines 18–26 (“We found the major axis… rotated ~45°…”)

Consider adding a supplementary figure showing this rotation; it will clarify the anatomical interpretation.

P13 – Lines 1–5 (ANESTHESIA RISK SECTION)

Important Comment:

This is an important clinical section, but currently descriptive.

Please add 1–2 practical guidance sentences—for example:

  • Recommended maximum needle advancement, or
  • How knowledge of canal length may reduce risk of globe penetration.

This will strengthen clinical applicability.(PDF annotation applied at the exact lines.)

P13 – Lines 10–20 (Regression models)

Discuss limitations of using moderate correlations (r ≈ 0.47–0.53).

Also note that findings apply mainly to mesocephalic European Shorthair cats.

P14 – Lines 3–10 (“The scarce literature… difficulty…”)

Rephrase to maintain a neutral scientific tone (avoid subjective “difficulty performing the work”).(PDF comment provided.)

  1. Conclusions (Page 15)

P15 – Lines 1–8

This section repeats the Abstract. Consider condensing and emphasizing:

— key findings,

— clinical application, and

— limitations/future work.

P15 – Lines 9–12 (“advancing a catheter may be unnecessary…”)

This is a strong clinical recommendation. Prefer cautionary phrasing unless supported by prospective clinical outcome data.(PDF note included.)

  1. References (Page 16)

P16 – Entire reference list. Several journal names appear abbreviated. MDPI requires full journal titles—please standardize.

Final Overall Comment

A useful and clinically relevant manuscript. With improvements in clarity, structure, figure annotation, and expanded clinical interpretation, this work will provide valuable reference data for feline dentistry and anesthesia.

Additional specific edits, textual corrections, and suggested rephrasing have been inserted directly into the annotated PDF for your convenience.

Comments on the Quality of English Language

The English is generally understandable, but several sentences are long or unclear, and some grammatical inconsistencies are present. A careful language revision is recommended to improve clarity, flow, and overall readability.

Author Response

Response to Reviewer 1 Comments

1. Summary

2. Questions for General Evaluation

Reviewer’s Evaluation

Response and Revisions

Does the introduction provide sufficient background and include all relevant references?

Can be improved

Introduction and methods were improved according to the suggestions. Please, see the below Point-by-point response to Comments and Suggestions for Authors

Is the research design appropriate?

Yes

Are the methods adequately described?

Can be improved

Are the results clearly presented?

Yes

Are the conclusions supported by the results?

Yes

Are all figures and tables clear and well-presented?

Yes

3. Point-by-point response to Comments and Suggestions for Authors

Comments 1: Simple Summary (Page 1) P1 – Lines 3–7 (“This study describes the European Shorthair Cats orbital…”) Please simplify long sentences for readability. The current text is dense. (See PDF annotation.)

Response 1: Thank you for pointing this out. We agree with this comment. Therefore, we have revised the sentence accordingly (page 1, lines 15-16)

o            Previous text: ‘This study describes the European Shorthair Cats orbital and infraorbital anatomy using simplified computed tomographic (CT) measurements to support safer locoregional anesthesia and surgical planning.’

o            Revised text: ‘This study describes the European Shorthair Cats’ orbital and infraorbital anatomy to support safer locoregional anesthesia and surgical planning.’ 

Comments 2: Simple Summary (Page 1) P1 – Line 10 (“using simplified computed tomographic measurements”) Clarify what is meant by “simplified”—simplified compared to which previous CT protocols? (PDF annotation included.)

Response 2: We thank the reviewer for this helpful comment. By “simplified computed tomographic measurements,” we refer to an approach that relies exclusively on direct linear measurements obtained from standard sagittal, transverse, and dorsal CT planes, without the need for multiplanar reformats, volumetric segmentation, or three-dimensional (3D) reconstructions, similar to the performed by Davis et al. (2021). To clarify this point, we have revised the Simple Summary. (Page 1, lines 16-19).

o            Previous text: ‘This study describes the European Shorthair Cats orbital and infraorbital anatomy using simplified computed tomographic (CT) measurements to support safer locoregional anesthesia and surgical planning.’

o            Revised text: ‘The objective was to provide reliable anatomical landmarks and reference values, using an approach based on linear measurements taken directly from sagittal, transverse and dorsal CT planes, instead of three-dimensional reconstructions.’

Comments 3: Abstract (Page 1) P1 – Lines 17–19 The sentence beginning “Despite its clinical importance…” repeats content from the Simple Summary. Consider condensing.

Response 3: We thank the reviewer for this observation. We agree that the sentence beginning “Despite its clinical importance…” partially repeated information already presented in the Simple Summary. To address this, the Abstract was revised by condensing this section and removing redundant wording. (Page 1, lines 28-30).

o            Previous text: ‘Despite its clinical importance, morphometric data for these regions are scarce, and existing studies typically relied on multiplanar or three-dimensional reconstructions of computed tomographic (CT) images, which require additional processing and expertise.’

o            Revised text: ‘Previous studies relied on multiplanar or three-dimensional reconstructions of computed tomographic (CT) images, requiring additional processing and expertise.’

Comments 4: Abstract (Page 1) P1 – Line 26 When reporting “Mean infraorbital canal length was 5.23 mm…”, please briefly mention its clinical significance (risk of globe penetration, block depth, etc.). (See PDF note.)

Response 4: We thank the reviewer for this valuable suggestion. We agree that reporting the mean infraorbital canal length benefits from an explicit statement of its clinical relevance. Accordingly, the Abstract was revised to briefly link the reported value to its clinical implications, namely the “limited safe depth for infraorbital/maxillary nerve blocks and the associated risk of globe penetration if instruments are advanced too far into the “infraorbital canal”. (Page 1, lines 39-41).

o            Added sentence: ‘This study contributes practical reference data to support the limited safe depth for infraorbital/maxillary nerve blocks and the associated risk of globe penetration if instruments are advanced too far into the infraorbital canal.’

Comments 5: Abstract (Page 1) P1 – Lines 29–34 The sentence on sex differences is lengthy; consider restructuring for clarity.

Response 5: We thank the reviewer for this observation. The sentence was restructured accordingly (Page 1, lines 37-38).

o            Previous text: ‘Male cats had larger infraorbital foramen dimensions than females. After adjustment for skull size, some differences maintained, reflecting a skull proportionally longer than wide in males.’

o            Revised text: ‘Males exhibited 0.42mm larger infraorbital major axis. After adjustment for skull size, only selected differences persisted, reflecting proportionally longer skulls in males.’

Comments 6: Introduction (Pages 2–3). P2 – Lines 6–20 The anatomy description is lengthy. Consider condensing and focusing more on the gap in feline morphometry. (PDF annotations highlight suggested trimming.)

Response 6: We agree with the comment. The anatomy description was shortened (Page 2, lines 50-60).

o            Previous text: ‘The maxillary nerve (cranial nerve V2, from trigeminal nerve) exits the skull through the foramen rotundum and exteriorizes in the rostral alar foramen at the pterygoid process of the sphenoid. Courses ventrally to the eyeball in the pterygopalatine fossa to finally enter the maxillary foramen. Along with its branch, the infraorbital nerve, which traverses the infraorbital canal, emerging through the infraorbital foramen, supply sensory innervation to the maxillary teeth, upper lip, and surrounding structures. The maxillary nerve is also related to the autonomous innervation of the lacrimal gland by carrying postganglionic parasympathetic fibres from the facial nerve (cranial nerve VII) via the pterygopalatine ganglion. The infraorbital canal is much shorter than the bony orbit floor in the cat, and the risk of globe penetration is higher [3]. Accurate localization of these landmarks is essential for safe infraorbital nerve blocks and for planning surgical approaches [4]. However, detailed morphometric descriptions of this region remain scarce, limiting the availability of reference values for clinical application.’

o            Revised text: ‘The maxillary nerve (branch of trigeminal nerve) exits the skull through the foramen rotundum and exteriorizes in the rostral alar foramen of the sphenoid. Courses ven-trally to eyeball in the pterygopalatine fossa, to finally enter the maxillary foramen. Along with its branch, the infraorbital nerve, which traverses the infraorbital canal, emerging through the infraorbital foramen, supplies sensory innervation to the maxil-lary teeth, upper lip, and surrounding structures. In cats, the infraorbital canal is much shorter than the bony orbit floor, and the risk of globe penetration is higher [3]. Accu-rate localization of these landmarks is essential for safe infraorbital nerve blocks and for planning surgical approaches [4]. However, detailed morphometric descriptions of this region remain scarce, limiting the availability of reference values for clinical ap-plication.’

Comments 7: Introduction (Pages 2–3). P2 – Lines 21–42 – Human morphometry section This section is useful but too long; condense to the essential points. Emphasize relevance to feline anatomy.

Response 7: We agree with your comment. The sentence was condensed to the essential points. (page 1, lines 61-68)

o            Previous text: ‘In humans, several cadaveric and imaging studies have described the infraorbital region in detail. Agthong et al. (2005) measured distances from the infraorbital fora-men to midline and to key anatomical landmarks, showing sex- and side-related differences [5]. Aggarwal et al. (2015) studied infraorbital foramen location, morphology, and its relationship to maxillary teeth, also reporting the frequency of accessory foramina [6]. Ercikti et al. (2017) correlated the foramen with palpable soft tissue land-marks to aid clinical localization [7]. Nam et al. (2017) mapped the infraorbital nerve, canal, and foramen relationships and documented accessory foramina [8]. Mar-tins-Júnior et al. (2017) examined the number, shape, and size of infraorbital and accessory foramina [9]. Fontolliet et al. (2019) used Cone Beam Computed Tomography (CBCT) to measure infraorbital canal dimensions, angles, and its relation to adjacent structures [10]. Polo et al. (2019) combined histology and computed tomography (CT) to demonstrate how accessory foramina affect infraorbital nerve branching [11]. Infraorbital nerve was also described as a surgically relevant landmark for the pterygopalatine fossa, cavernous sinus, and anterolateral skull base in endoscopic transmaxillary approaches in human neurosurgery [12] (Elhadi et al., 2016). Collectively, these studies provide a robust anatomical framework but are limited to humans.’

o            Revised text: ‘In humans, several cadaveric and imaging studies have described in detail the in-fraorbital region, foramen, canal and nerve, as well as their relationships to adjacent structures as the maxillary teeth, and palpable soft tissue landmarks to aid clinical lo-calization [5-10]. Agthong et al. (2005) measured distances from the infraorbital fora-men to midline, showing sex-related differences [11]. Accessory foramina were de-scribed [5, 7, 8]. Infraorbital nerve was reported as a relevant landmark for the ptery-gopalatine fossa, cavernous sinus, and anterolateral skull base in neurosurgery [12]. Collectively, these studies provide a robust anatomical framework but are limited to humans.’

Comments 8: Introduction (Pages 2–3). P3 – Line 9 (“The aims of this study were…”) Please explicitly state the novelty:

— Linear-plane measurements directly from CT planes vs.

— 3D reconstructions in previous studies (e.g., Davis et al.). (PDF comment added.)

Response 8: We agree with your comment. (pages 2-3, lines 84-91)

o            Previous text: ‘The aims of this study were to analyse retrospectively clinical scans from Euro-pean Shorthair Cats, using a simplified approach based on linear measurements taken directly from sagittal, transverse and dorsal CT planes, to assess repeatability, report infraorbital and cranial ratios, and develop regression models to predict infraorbital dimensions from basic skull measures. By proposing this simplified and clinic-ready workflow, our study contributes practical reference data and provides a foundation for standardization of feline infraorbital morphometry.’

o            Revised text: ‘The aims of this study were to analyze retrospectively clinical scans from Euro-pean Shorthair Cats, using a simplified approach based on linear measurements taken directly from sagittal, transverse and dorsal CT planes, instead of 3D reconstructions previously described [15], to assess repeatability, report infraorbital and skull ratios, and develop regression models to predict infraorbital dimensions from basic skull measures. By proposing this simplified and clinic-ready workflow, our study contributes practical reference data and provides a foundation for standardization of feline infraorbital morphometry.’

Comments 9: Materials and Methods (Pages 4–5). P4 – Line 4 (“older than 9 months”)

Provide justification or reference for skeletal maturity at 9 months.

Response 9: Thank you for raising this important point. This was a typing error of our part. European Shorthair cats older than 24 months were included to ensure skeletal maturity. All animals in our study aged between 4 and 17 years, with a mean age of 10.14 ± 3.73 years. (page 3, lines 97-98)

o            Previous text: ‘European Shorthair cats older than 9 months were included to ensure skeletal maturity. Exclusion criteria were cranial abnormalities or poor image quality.’

o            Revised text: ‘European Shorthair cats older than 24 months were included to ensure skeletal maturity [21, 22].’ 

We had new references regarding this point:

21.         Smith RN. Fusion of ossification centres in the cat. Journal of Small Animal Practice 1969, 10, 523–530.

22.         Miranda, F. G., Souza, I. P., Viegas, F. M., Megda, T. T., Nepomuceno, A. C., Tôrres, R. C., Rezende, C. M. Radiographic study of the development of the pelvis and hip and the femorotibial joints in domestic cats. Journal of Feline Medicine and Surgery, 2020, 22, 476–483. https://doi.org/10.1177/1098612X19854809

Comments 10: Materials and Methods (Pages 4–5). P4 – Lines 12–15 (“Images were acquired at ~120 kVp…”) Specify slice interval (spacing), not just thickness.

Response 10: Thank you for this observation. The slice interval was added. (page 2, lines 104-105)

o            Previous text: ‘Regarding the imaging protocol, CT scans were performed with a 16-slice helical scanner (Aquilion Prime SP, Canon). Cats were positioned in ventral recumbency with the head aligned and immobilized. Images were acquired at ~120 kVp, 120 mAs, with a slice thickness of 1 mm.’

o            Revised text: ‘Regarding the imaging protocol, CT scans were performed with a 16-slice helical scanner (Aquilion Prime SP, Canon). Cats were positioned in ventral recumbency with their head aligned and immobilized. Transverse slices of the head were acquired at ~120 kVp, 120 mAs, with a slice thickness of 1 mm, and a slice interval of 0,8 mm.’ 

Comments 11: Materials and Methods (Pages 4–5). P4 – Lines 20–30 (“The sagittal plane… dorsal plane…”) Clarify how rotational misalignment of the head was corrected before measurements. This is important for reproducibility.

Response 11: Thank you for raising this important point. The text was modified to clarify the procedure. (page 3, lines 107-112)

o            Revised text: ‘In order to perform the linear measurements described in Table 1, firstly, rota-tional misalignment of the head was corrected by aligning the images in the three planes, as demonstrated in Figure 1. The sagittal plane (Figure 1-A) was aligned at the level of the nasal septum and the interincisive midline; the transverse plane (Figure 1-B) was oriented perpendicular to the hard palate; and the dorsal plane (Figure 1-C) was parallel to the hard palate.’

Comments 12: Materials and Methods (Pages 4–5). P4 – Line 33 (“two measurements of each parameter were performed”) Justify why two measurements were chosen instead of three (common in morphometric studies). (PDF note included.)

Response 12: A preliminary study and prior training were conducted, to establish anatomical landmarks that would allow for the repeatability of the measurement methodology by authors JFR, ARS, SAP, and since the repeatability results were good, only two measurement sessions were performed. (page 3, lines 116-129)

o            Previous text: ‘Linear morphometric parameters (Table 1) were obtained directly from transverse 122 CT images using Horos™ DICOM software with bone filters. Recorded variables included infraorbital foramen major axis (Figure 2), minor axis (Figure 3) and length (Figure 4), distance between infraorbital foramina (DIF) (Figure 5), orbital height and width (Figure 6 and Figure 7), zygomatic arch width (Figure 8), skull width and length (Figure 9). Ratios were calculated to normalize for skull size. In order to reduce the analysis margin of error, two measurements of each studied parameter were performed. The measurements were performed by the same operator (to reduce interpersonal errors), and each measurement of each parameter was performed at different times, in order to reduce intrapersonal errors. Then, the arithmetic mean of the measurements was calculated.’

o            Revised text: ‘Linear morphometric parameters (Table 1) were obtained directly from CT images using Horos™ v3.3.6.dmg DICOM software with bone filters. Prior to measurement, all images were calibrated from pixels to millimeters. A preliminary study and prior training were conducted, to establish anatomical landmarks that would allow for the repeatability of the measurement methodology by JFR, ARS, SAP. Recorded variables included infraorbital foramen major axis (Figure 2), minor axis (Figure 3) and length (Figure 4), distance between infraorbital foramina (DIF) (Figure 5), orbital height and width (Figure 6 and Figure 7), zygomatic arch width (Figure 8), skull width and length (Figure 9). Ratios were calculated to normalize for skull size. As preliminary results revealed adequate repeatability of measurements, and in order to reduce the analysis margin of error, two measurements of each studied parameter were performed. The measurements were performed by the same operator (to reduce interpersonal errors), and each measurement of each parameter was performed at different times, in order to reduce intrapersonal errors. Then, the arithmetic mean of the measurements was calculated.’

Comments 13: Materials and Methods (Pages 4–5). P5 – Figures 2–5 Add orientation labels (L/R, dorsal/ventral) to improve interpretation. (Annotated on PDF.)

Response 13: We thank the reviewer for this helpful suggestion. Orientation labels indicating left/right (L/R), dorsal/ventral (D/V) and cranial/caudal (Cr/Cd) directions have now been added to Figures 1 to 10 to improve image interpretation and anatomical clarity. These additions facilitate spatial orientation for readers.

Comments 14: Results (Pages 7–11). P7 – Line 3 (“Twenty-four European Shorthair cats…”)

Include neuter status if available. Sex hormones can influence cranial morphology.

Response 14: Thank you for this suggestion. Neuter status was added. The sentence was moved for the material and methods by suggestion of reviewer 2 (page 3, lines 99-101)

o            Previous text: ‘Twenty-four European Shorthairs cats of both sexes (12 females and 12 males), aged between 4 and 17 years, with a mean age of 10.14 ± 3.73 years were included. Their mean weight was 4.67 ± 1.31 kg, ranging from 3.25 kg to 7.65 kg.’

o            Revised text: ‘Twenty-four CT scans of neutered feline heads, of both sexes (12 females and 12 males), aged between 4 and 17 years (mean age ± standard deviation (SD) of 10.14 ± 3.73 years) were evaluated. Their mean weight was 4.67 ± 1.31 kg, ranging from 3.25 kg to 7.65 kg.’

Comments 15: Results (Pages 7–11). P7 – Lines 12–15 (“repeatability analysis…”)

Explain what ICC ≥ 0.76 indicates (good to excellent repeatability). (PDF annotation added.)

Response 15: An ICC of 1.0 indicates perfect agreement, and an ICC of 0.0 indicates no agreement. Lower limits of the 95% confidence interval (CI) ≥ 0.75 indicate a strong association

between the two measurements (Lee et al., 1989; Zou, 2012). Regarding repeatability analysis, new sentences and reference were added for a better explanation. (Pages 3-4, lines 134-141).

o            Revised text in material and methods: ‘Intra-observer repeatability was assessed using Bland–Altman analysis and intra-class correlation coefficients (ICC) [23, 24]. In the Bland–Altman analysis, the 95% limits of agreement (LA) were calculated as the mean difference ± 1.96 SD, where SD is the standard deviation of all individual differences. When the 95% confidence interval (CI) of the mean differences includes the 0, measurements are considered to be in agreement; and when the 95% lower and upper limits of agreement are small, measurements are considered to be equivalent, with adequate intra-observer agreement [23]. An ICC of 1.0 indicates perfect agreement, and an ICC of 0.0 indicates no agreement [24, 25].‘

o            Revised text in the results: ‘All the 95% CI of the mean differences included the 0, meaning that measurements are considered to be in agreement; and the intervals between 95% lower and upper limits of agreement are small, so, measurements are considered to be equivalent, with adequate intra-observer agreement [23]. Regarding the ICC, lower limits of the 95% CI ≥ 0.75 indicate a strong association between the two measurements [24, 25]. The ICC for intraobserver repeatability were all ≥ 0.75, with the lowest value (0.76) observed for Orbital width, indicating statistically adequate repeatability.’

Reference added: Zou G. Y. (2012). Sample size formulas for estimating intraclass correlation coefficients with precision and assurance. Statistics in medicine, 31(29), 3972–3981. https://doi.org/10.1002/sim.5466

Comments 16: Results (Pages 7–11). P7 – Lines 18–30 (Correlation and regression) Please report R² values in addition to r values to quantify predictive strength. Tables 2–4 (Pages 8–9) Some ratio formulas (e.g., IFMA × 20) are unclear. Please explain the rationale for numerical multipliers.

PDF: (Page 7) Please provide effect sizes (Cohen’s d) to quantify magnitude of sex differences.

Response 16: This information was added to the manuscript (page 7, lines 183-192, and Tables 5-6).

Ratios were calculated to normalize for skull size. However, in a preliminary analysis, we observed that Skull length (SL) Mean ± Standard deviation (Range) values of 91.55 ± 5.68 (81.30-110.00) mm were ~20 superior to Infraorbital foramen major axis (IFMA) values of 3.82 ± 0.68 (2.56-5.58) mm, so we decided to multiply by 20 the IFMA to simplify interpretation. A sentence was added to the discussion.

o            Previous text: ‘A correlation was found between skull width and infraorbital canal length (r = 0.476, p=0,001). From the linear regression analysis between the length of the infraorbital canal and the width of the skull we obtained the regression equation: y = 0.0739x + 0.7427, where y = length of the infraorbital canal and x = width of the skull (r2 = 0,227).

Also, a moderate positive correlation was observed between skull length and infraorbital foramen minor axis (r = 0.531, p<0,001). From the linear regression analysis between the length of the skull and the minor axis of the infraorbital foramen we obtained the regression equation: y = 0.0499x - 1.772, where y = minor axis of the infraorbital foramen and x = length of the skull (r2 = 0,282).’

o            Revised text: ‘Pearson correlation measures the strength and direction of linear relationships between variables, ranging from -1.0 to +1.0, where weak (0.0 to 0.3), moderate (0.3 to 0.7), and etrong (0.7 to 1.0) correlations can be found. A moderate positive correlation was found between Skull width (SW) and Infraorbital canal length (ICL) (r = 0.476, p=0.001). From the linear regression analysis between the ICL and the SW we obtained the regression equation: y = 0.0739x + 0.7427, where y = ICL and x = SW (r2 = 0.227).

Also, a moderate positive correlation was observed between Skull length (SL) and Infraorbital foramen minor axis (IFmA) (r = 0.531, p<0,001). From the linear regression analysis between the SL and the IFmA we obtained the regression equation: y = 0.0499x - 1.772, where y = IFmA and x = SL (r2 = 0.282).’

o            Added sentence in the discussion Pages 12-13, lines 266-271: ‘In a preliminary analysis, we observed that Skull length (SL) values (91.55 ± 5.68 mm) were ~20 superior to Infraorbital foramen major axis (IFMA) values (3.82 ± 0.68 mm), so we decided to multiply by 20 the IFMA to simplify the interpretation. Regarding the ratio (Infraorbital foramen minor axis × 2) / Skull width [(IFmA×2)/SW], we multiplied by two the IFmA, as SW includes the left and right IFmA.’

Comments 17: Discussion (Pages 12–14). P12 – Lines 1–8 (Opening paragraph)

This paragraph restates methodological details; consider focusing more on interpretation and comparison.

Response 17: The manuscript has been revised to address the reviewer’s suggestion, with the Discussion updated to emphasize interpretation and comparison rather than methodological repetition. (page 12, lines 219-231).

o            Previous text: ‘This study provides anatomical landmarks for the European Shorthair Cats infra-orbital region and comparable morphometric values in healthy domestic cats, which may be used as reference and contribute to the comprehension of the infraorbital region and its relations to the orbit and pterygopalatine fossa. Using a simplified approach, based on linear measurements taken directly from sagittal, transverse and dorsal CT planes, accurate data obtained may be used in future comparisons with other data, for planning surgical approaches and anesthetic nerve block procedures, and differs from previous by using plain CT images rather than multiplanar or 3D reconstructions [15].’

o            Revised text: ‘This study provides reference morphometric data for the infraorbital region of European Shorthair cats while critically evaluating the use of simplified CT-based measurements in feline craniofacial anatomy. Unlike previous studies that predomi-nantly relied on multiplanar reformatting or three-dimensional reconstructions, the present work demonstrates that linear measurements obtained directly from standard sagittal, transverse, and dorsal CT planes are sufficient to characterize clinically rele-vant infraorbital and orbital landmarks. The morphometric values reported here are largely consistent with those described in earlier CT-based investigations, although minor differences were observed, likely reflecting methodological variation, skull con-formation, and sample characteristics [15]. These findings suggest that increased methodological complexity does not necessarily confer greater anatomical precision and underscore the importance of balancing accuracy with clinical feasibility when translating imaging-based morphometry into routine veterinary practice.’

Comments 18: Discussion (Pages 12–14). P12 – Lines 18–26 (“We found the major axis… rotated ~45°…”). Consider adding a supplementary figure showing this rotation; it will clarify the anatomical interpretation.

Response 18: We thank the reviewer for this suggestion. The orientation of the infraorbital foramen major axis is already illustrated in Figure 2, where its oblique orientation relative to the mid-sagittal plane can be clearly observed. To avoid overinterpretation, we have revised the text by removing the approximate numerical value (45°) and now describe the major axis as obliquely oriented in relation to the median plane. We therefore consider that an additional supplementary figure is not necessary, as this anatomical feature is adequately illustrated in the main figure of the manuscript. (page 12, lines 243-247).

o            Previous text: ‘We found the major axis of the infraorbital foramen to be rotated ~45° relative to the vertical in European Shorthair Cats, unlike what has been described in dogs, and, for that reason, naming it total vertical height of the infraorbital foramen seemed inaccurate to us, so we rather named it the Infraorbital foramen major axis (IFMA) and the Infraorbital foramen minor axis (IFmA) [2].’

o            Revised text: ‘We found the major axis of the infraorbital foramen to have an oblique orientation relative to the mid-sagittal plane (Figure 2), unlike what has been described in dogs, and, for that reason, naming it total vertical height of the infraorbital foramen seemed inaccurate to us, so we rather named it the Infraorbital foramen major axis (IFMA) and the Infraorbital foramen minor axis (IFmA) [2].’

Comments 19: Discussion (Pages 12–14). P13 – Lines 1–5 (ANESTHESIA RISK SECTION)

Important Comment:

This is an important clinical section, but currently descriptive.

Please add 1–2 practical guidance sentences—for example:

•            Recommended maximum needle advancement, or

•            How knowledge of canal length may reduce risk of globe penetration.

This will strengthen clinical applicability. (PDF annotation applied at the exact lines.)

Response 19: The manuscript has been revised to address the reviewer’s suggestion, with the Discussion updated to emphasize clinical applicability. (page 14, lines 354-358).

o            Added sentence: ‘From an anatomical perspective, these results suggest that deep needle or catheter advancement into the infraorbital canal may not be necessary to achieve effective re-gional anesthesia and may increase the risk of iatrogenic injury. However, this inter-pretation is based on morphometric data rather than prospective clinical outcome studies and should therefore be applied with appropriate caution.’

Comments 20: Discussion (Pages 12–14). P13 – Lines 10–20 (Regression models)

Discuss limitations of using moderate correlations (r ≈ 0.47–0.53).

Also note that findings apply mainly to mesocephalic European Shorthair cats.

Response 20: The Discussion has been revised to acknowledge the moderate strength of the correlations and to clarify the limitations of the regression models. (page 13, lines 293-303).

o            Previous text: ‘To this end, we calculated the correlation coefficients and found a significant cor-relation between the length of the infraorbital canal and skull width of 0.476, and be-tween the minor axis of the infraorbital orifice and skull length of 0.531. From the regression analysis between the length of the infraorbital canal and the width of the skull, we obtained the regression line: y = 0.0739x + 0.7427. Thus, we can estimate the length of the infraorbital canal from the measurement of the width of the skull. From the regression analysis between the minor axis of the infraorbital canal and the length of the skull, we obtained the regression equation: y = 0.0499x - 1.772. Thus, in the future, we can estimate the minor axis of the infraorbital canal by knowing the length of the skull.’

o            Revised text: ‘From the regression analysis between the ICL and the SW, we obtained the regression line: y = 0.0739x + 0.7427; and from the regression analysis between the Infraorbital fo-ramen minor axis (IFmA) and Skull length (SL), the regression equation: y = 0.0499x - 1.772. Although the correlations between Infraorbital canal length (ICL) and Skull width (SW), and between Infraorbital foramen minor axis (IFmA) and Skull length (SL), were statistically significant, their magnitude was moderate (r ≈ 0.47–0.53). This indicates that these regression models explain only a limited proportion of the ob-served anatomical variability and should therefore be interpreted with caution. Consequently, the proposed equations are not intended to provide precise predictions at an individual level, but rather to offer approximate estimations that may assist clinical decision-making when advanced imaging is unavailable.’

Comments 21: Discussion (Pages 12–14). P14 – Lines 3–10 (“The scarce literature… difficulty…”) Rephrase to maintain a neutral scientific tone (avoid subjective “difficulty performing the work”).(PDF comment provided.)

Response 21: We thank the reviewer for this comment. The revised wording now emphasizes the limited availability of published data on this anatomical region. (page 14, lines 333-340).

o            Previous text: ‘The scarce literature on this anatomical region, especially in felines, added some difficulty in carrying out this work. For this reason, we consider the results obtained in this study to be a contribution to the characterization of the orbital and infraorbital re-gions in mesocephalic cats, which may add the planning of safer procedures as the administration of anesthetics in locoregional blocks and surgical approaches.’

o            Revised text: ‘The available literature addressing the orbital and infraorbital regions in the cat remains limited, particularly with respect to detailed morphometric characterization based on imaging studies. As a result, reference data for this anatomical area are still scarce, which may constrain direct comparisons across studies. Within this context, despite the relatively small sample size (n=24) and the non-random clinical scans study, the present work contributes additional imaging-based morphometric information for mesocephalic cats and complements previous descriptions of feline cranial anatomy, providing a basis for future comparative and clinically oriented investigations.’

Comments 22: Conclusions (Page 15). P15 – Lines 1–8

This section repeats the Abstract. Consider condensing and emphasizing:

— key findings,

— clinical application, and

— limitations/future work.

Response 22: We thank the reviewer for this comment. The Conclusions section has been revised and condensed to avoid repetition of the Abstract. The revised version now emphasizes the main findings, their clinical relevance, and the principal limitations of the study, while also outlining directions for future research. (page 14, lines 349-365).

o            Previous text: ‘This study provides the first set of reference values for the infraorbital region of European Shorthair Cats using sagittal, transverse and dorsal CT planes, demonstrating that simplified measurements are repeatable and clinically applicable. The infraorbital canal length in our sample ranged from 4.42 to 6.33, with a mean of 5.23 mm, consistent with previous estimates for mesocephalic cats. Clinically, this finding supports the concept that advancing a catheter into the infraorbital canal may be unnecessary, and deposition at the infraorbital foramen alone appears sufficient to achieve maxillary nerve coverage, while avoiding complications such as ocular trauma associated with catheter placement and dye migration intracranially via the alar foramina. This also highlights the importance of calculate the minimum effective injectate volume and tailoring block technique to the cat specific anatomy.

By introducing a straightforward, clinic-ready method for infraorbital morphometry, our study complements previous 3D reconstruction studies and extends the general feline skull morphometry described [15,16]. Together, these contributions enrich the anatomical framework necessary for safer and more effective infraorbital and maxillary nerve blocks in cats, while laying the groundwork for standardization of imaging-based protocols in feline anaesthesia and surgery. By proposing this simplified and clinic-ready workflow, based on linear measurements taken directly from sagittal, transverse and dorsal CT planes, our study contributes practical reference data and provides a foundation for standardization of feline infraorbital morphometry.’

o            Revised text: ‘This study provides CT-based reference values for the infraorbital region of neutered European Shorthair cats using a simplified, repeatable measurement approach. The findings confirm that the infraorbital canal is short and lies in close proximity to the orbit, underscoring the anatomical relevance of this region for infraorbital and maxillary nerve block techniques.

From an anatomical perspective, these results suggest that deep needle or catheter advancement into the infraorbital canal may not be necessary to achieve effective regional anesthesia and may increase the risk of iatrogenic injury. However, this interpretation is based on morphometric data rather than prospective clinical outcome studies and should therefore be applied with appropriate caution.

The simplified methodology proposed may facilitate broader clinical use of CT-derived morphometry without the need for advanced image reconstruction techniques. Nevertheless, the results are limited to mesocephalic European Shorthair cats, and extrapolation to other breeds or skull conformations should be undertaken care-fully. Future studies incorporating larger and more diverse feline populations, as well as prospective clinical outcome data, are warranted to further refine anatomical guide-lines for safer locoregional anesthetic techniques.’

Comments 23: Conclusions (Page 15). P15 – Lines 9–12 (“advancing a catheter may be unnecessary…”)

This is a strong clinical recommendation. Prefer cautionary phrasing unless supported by prospective clinical outcome data. (PDF note included.)

Response 23: We agree with the reviewer’s observation. The Conclusions section has been revised to adopt more cautious wording regarding catheter advancement. (page 14, lines 354-358).

o            Previous text: ‘Clinically, this finding supports the concept that advancing a catheter into the infraorbital canal may be unnecessary, and deposition at the infraorbital foramen alone appears sufficient to achieve maxillary nerve coverage, while avoiding complications such as ocular trauma associated with catheter placement and dye migration intracranially via the alar foramina.’

o            Revised text: ‘From an anatomical perspective, these results suggest that deep needle or catheter advancement into the infraorbital canal may not be necessary to achieve effective regional anesthesia and may increase the risk of iatrogenic injury. However, this interpretation is based on morphometric data rather than prospective clinical outcome studies and should therefore be applied with appropriate caution.’

Comments 24: References (Page 16). P16 – Entire reference list. Several journal names appear abbreviated. MDPI requires full journal titles—please standardize.

Response 24: We thank the reviewer for this comment. The References section has been carefully revised to ensure that all journal titles are written in full, in accordance with MDPI guidelines.

Comments 25: Final Overall Comment. A useful and clinically relevant manuscript. With improvements in clarity, structure, figure annotation, and expanded clinical interpretation, this work will provide valuable reference data for feline dentistry and anesthesia. Additional specific edits, textual corrections, and suggested rephrasing have been inserted directly into the annotated PDF for your convenience.

Response 25: We thank you again for the thoughtful and constructive comments, which have helped to clarify the methods, contextualize the results, and improve the overall quality of the manuscript. We have revised the manuscript, and now we believe it meets all the criteria laid down.

4. Response to Comments on the Quality of English Language

Point 1: Comments on the Quality of English Language. The English is generally understandable, but several sentences are long or unclear, and some grammatical inconsistencies are present. A careful language revision is recommended to improve clarity, flow, and overall readability.

Response 1: Thank you for your feedback. The entire manuscript was revised by a native speaker.

December 29, 2025

Reviewer 2 Report

Comments and Suggestions for Authors

This manuscript presents a retrospective computed tomography (CT) study characterizing the orbital and infraorbital regions in 24 European Shorthair cats, focusing on linear measurements from standard sagittal, transverse, and dorsal planes. The authors aim to provide reference values and anatomical landmarks to enhance safety in locoregional anesthesia (e.g., infraorbital and maxillary nerve blocks) and surgical planning in feline dentistry and maxillofacial procedures. Key findings include repeatable measurements, correlations between infraorbital canal dimensions and skull parameters (with regression models), and sex-specific differences that partially persist after normalization for skull size.

The study is clinically relevant, addressing a gap in feline-specific morphometric data, where prior work often relies on more complex 3D reconstructions or focuses on other species. The simplified approach is a strength, making it accessible for clinical use without advanced software. Below are some minor points worth considering when revising:

Abstract is clear but could quantify sex differences (e.g., "males had 0.42 mm larger IFMA").

Lines 21, 35: "Infraorbital measurements/dimensions" - specify whether this refers to the infraorbital foramen or the infraorbital canal.

Line 90: "European Shorthair skulls" citation [16] is from the authors—note if it's the same cohort.

Line 136: Mention if normality was tested (e.g., Shapiro-Wilk) before Pearson/t-tests.

Lines 191-193: Consider moving to Materials and Methods section.

Author Response

Response to Reviewer 2 Comments

1. Summary

2. Questions for General Evaluation

Reviewer’s Evaluation

Response and Revisions

Does the introduction provide sufficient background and include all relevant references?

Can be improved

Introduction and research design were improved accordingly. Please, see the below Point-by-point response to Comments and Suggestions for Authors

Is the research design appropriate?

Can be improved

Are the methods adequately described?

Yes

Are the results clearly presented?

Yes

Are the conclusions supported by the results?

Yes

Are all figures and tables clear and well-presented?

Yes

3. Point-by-point response to Comments and Suggestions for Authors

Comments 1: Abstract is clear but could quantify sex differences (e.g., "males had 0.42 mm larger IFMA").

Response 2: We appreciate this suggestion. Information regarding sex differences was added to the abstract (pages 1, lines 35-37)

o            Revised text: ‘Significant differences were observed between mean infraorbital canal length in fe-males (5.02±0.42) and males (5.46±0.46), and skull length in females (88.06±3.16) and males (94.54±4.52). Males exhibited 0.42mm larger infraorbital major axis.’

Comments 2: Line 90: "European Shorthair skulls" citation [16] is from the authors—note if it's the same cohort.

Response 2: Thank you for raising this important point. It was a different cohort (pages 2, lines 78-80).

o            Previous text: ‘Ramos et al. (2021) characterized European Shorthair skulls morphometrically using CT, though without focusing on the infraorbital region [16].’

o            Revised text: ‘Ramos et al. (2021) characterized a different cohort of European Shorthair skulls morphometrically using CT, though without focusing on the infraorbital region [16].’

Comments 3: Line 136: Mention if normality was tested (e.g., Shapiro-Wilk) before Pearson/t-tests.

Response 3: Yes. We have performed the Shapiro-Wilk test before performing the Pearson/t-tests.

o            New sentence added in the material and methods session (pages 3, lines 131-132): ‘The normality of data was assessed using the Shapiro-Wilk test.’

Comments 4: Lines 191-193: Consider moving to Materials and Methods section.

Response 4: We agree with your suggestion. The sentence was moved to Materials and Methods section (pages 3, lines 99-101):

o            Previous text: Twenty-four neutered European Shorthairs cats of both sexes (12 females and 12 males), aged between 4 and 17 years, with a mean age of 10.14 ± 3.73 years were included. Their mean weight was 4.67 ± 1.31 kg, ranging from 3.25 kg to 7.65 kg.’

o            Revised text: ‘Twenty-four CT scans of neutered feline heads, of both sexes (12 females and 12 males), aged between 4 and 17 years (mean age ± standard deviation (SD) of 10.14 ± 3.73 years) were evaluated. Their mean weight was 4.67 ± 1.31 kg, ranging from 3.25 kg to 7.65 kg.’

4. Response to Comments on the Quality of English Language

Point 1: The English is fine and does not require any improvement.

Response 1: Thank you for your feedback. The entire manuscript was revised by a native speaker.

Once again, we sincerely thank you for the constructive and insightful suggestions, which have significantly strengthened the conceptual and methodological clarity of our manuscript.

December 29, 2025
